# Cardiovascular safety of celecoxib in rheumatoid arthritis and osteoarthritis patients: A systematic review and meta-analysis

Bai-Ru Cheng[1], Jia-Qi Chen[2], Xiao-Wen Zhang[3], Qin-Yang Gao[1], Wei-Hong Li[4‡], Li-Jiao Yan[3‡], Yu-Qiao Zhang[2], Chang-Jiang Wu[5], Jing-Li Xing[3], Jian-Ping Liu[3]*

1 The First School of Clinical Medicine (Dongzhimen Hospital), Beijing University of Chinese Medicine, Beijing, China, 2 Clinical College (China-Japan Friendship Hospital), Beijing University of Chinese Medicine, Beijing, China, 3 Centre for Evidence-Based Chinese Medicine, Beijing University of Chinese Medicine, Beijing, China, 4 School of Nursing, Beijing University of Chinese Medicine, Beijing, China, 5 The Second School of Clinical Medicine (Dongfang Hospital), Beijing University of Chinese Medicine, Beijing, China

☯ These authors contributed equally to this work.
‡ These authors also contributed equally to this work.
* liujp@bucm.edu.cn

**Data Availability Statement:** All relevant data are within the paper and its Supporting Information files.

## Abstract

### Objective

To assess the cardiovascular safety of celecoxib compared to non-selective non-steroid anti-inflammatory drugs or placebo.

### Methods

We included randomized controlled trials of oral celecoxib compared with a non-selective NSAID or placebo in rheumatoid arthritis and osteoarthritis patients. We conducted searches in EMBASE, Cochrane CENTRAL, MEDLINE, China National Knowledge Infra-structure, VIP, Wanfang, and Chinese Biomedical Literature Database. Study selection and data extraction were done by two authors independently. The risk of bias was assessed using Cochrane's risk-of-bias Tool for Randomized Trials. The effect size was presented as a risk ratio with their 95% confidence interval.

### Results

Until July 22nd, 2021, our search identified 6279 records from which, after exclusions, 21 trials were included in the meta-analysis. The overall pooled risk ratio for Antiplatelet Trialists Collaboration cardiovascular events for celecoxib compared with any non-selective non-steroid anti-inflammatory drugs was 0.89 (95% confidence interval: 0.80–1.00). The pooled risk ratio for all-cause mortality for celecoxib compared with non-selective non-steroid anti-inflammatory drugs was 0.81 (95% confidence interval: 0.66–0.98). The cardiovascular mortality rate of celecoxib was lower than non-selective non-steroid anti-inflammatory drugs (risk ratio: 0.75, 95% confidence interval: 0.57–0.99). There was no significant difference

**Funding:** This review was supported by the National Natural Science Foundation project (No. 81830115), and partially supported by the NCCIH grant (AT001293 with sub-award No. 020468C).

**Competing interests:** The authors have declared that no competing interests exist.

between celecoxib and non-selective non-steroid anti-inflammatory drugs or placebo in the risk of other cardiovascular events.

## Conclusion

Celecoxib is relatively safe in rheumatoid arthritis and osteoarthritis patients, independent of dose or duration. But it remains uncertain whether this would remain the same in patients treated with aspirin and patients with established cardiovascular diseases.

## Introduction

Rheumatoid arthritis (RA) is a chronic, inflammatory disease of which pathological mechanism remains unclear [1]. Osteoarthritis (OA) is the most common joint disease worldwide [2]. Compared with the general population, the mortality rate among rheumatoid arthritis patients is higher, which is largely attributable to cardiovascular disease, particularly (fatal and non-fatal) myocardial infarction due to coronary atherosclerosis [3]. Risks of both myocardial infarctions and strokes are amplified in individuals with rheumatoid arthritis, which may be the result of inflammation-associated vascular damage [4, 5]. Likewise, the risks of heart failure and ischemic heart disease increase in osteoarthritis patients [6]. It is necessary to control cardiovascular events, especially fatal cardiovascular events for rheumatoid arthritis and osteoarthritis patients.

Nonsteroidal anti-inflammatory drugs (NSAIDs) are often prescribed to relieve arthritis symptoms. Cyclo-oxygenase (COX)-2 inhibitors (coxibs), which were developed as alternative analgesics to minimize upper gastrointestinal toxicity of non-selective NSAIDs (nsNSAIDs), were believed to reduce cardiovascular risks [7]. Celecoxib (Celebrex®) was the first coxib introduced into clinical practice. At its recommended doses of 200 or 400 mg/day, it is as effective for symptomatic treatment as conventional NSAIDs and some other coxibs [8] and can significantly reduce upper gastrointestinal events [9]. However, the subsequent coxib, rofecoxib, was found to increase cardiovascular events, which led to its worldwide withdrawal in 2004 [10]. Another coxib, valdecoxib (Bextra®), was also withdrawn from the market in 2005 for its cardiovascular toxicity and serious skin reactions [11]. The withdrawal raised concern about the cardiovascular safety of coxibs. As a result, the use of many coxibs was restricted and, all coxibs and nonselective NSAIDs were recommended to be used with caution in patients with older age or established cardiovascular disease [12, 13].

Cardiovascular safety on NSAIDs is highly controversial. Although it has been confirmed that some NSAIDs are associated with a higher risk of cardiovascular events, the cardiovascular safety profile varies widely. Coxibs were associated with increased risks of myocardial infarction when compared against placebo or non-selective NSAIDs [14]. Salpeter et al. suggested nonselective NSAIDs had no significant effect on cardiovascular events or death in trials of joint disease and Alzheimer's disease, and there was no significant adverse or cardioprotective effect of naproxen [15]. Kearny et al. found diclofenac and ibuprofen were associated with higher cardiovascular risk, while naproxen was not [16]. A meta-analysis by Coxib and traditional NSAID Trialists' Collaboration indicated that the vascular risks of high-dose diclofenac, and possibly ibuprofen, were comparable to coxibs, whereas high-dose naproxen was associated with less vascular risk than other NSAIDs [7]. In OA patients, the risk of heart failure in the coxib group remained significant even when rofecoxib was removed from the analysis [17].

Also, the cardiovascular safety of celecoxib compared to other NSAIDs remains uncertain, especially among patients with established cardiovascular diseases [18]. One Cochrane review [19] suggested the risk of bias might be the reason for uncertainty about the rate of cardiovascular events between celecoxib and nsNSAIDs. Another factor was that most trials were small and short-term. It is also because mechanisms of the cardiovascular side effects are still controversial. One hypothesis is that coxibs have an impact on vasoactive endothelium-derived factors, particularly via inhibiting prostaglandin synthesis, which is important for the regulation of vascular tone and sodium excretion and may influence blood pressure, but it is not for certain yet. This systematic review and meta-analysis aimed to assess the cardiovascular safety of celecoxib compared to non-selective nonsteroidal anti-inflammatory drugs and placebo.

## Methods

Our protocol was published on PROSPERO [CRD42020179936] with the title *Cardiovascular Safety of Celecoxib for Rheumatoid Arthritis and Osteoarthritis*: *A Systematic Review and Meta-Analysis*.

### Search strategy

Publications were retrieved using computerized searches by EMBASE, CENTRAL, MEDLINE, CNKI, VIP, Wanfang, and the Chinese Biomedical Literature Database. No date or language limits were set. A re-run was done before the final analyses. The last search date was July 22$^{nd}$, 2021.

### Inclusion criteria

Patients with rheumatoid arthritis (diagnosed according to ACR 1987 criteria [20] or ACR/EULAR 2010 criteria [21]) or osteoarthritis (diagnosed according to ACR guidelines [22, 23]) were included, while patients with other rheumatic diseases such as systemic lupus erythematosus or Sjogren's Syndromes were excluded. Only RCTs comparing celecoxib at any dose to non-selective NSAIDs or placebo were included.

### Outcome measures

Primary outcomes: 1. all-cause mortality; 2. cardiovascular mortality; 3. fatal and non-fatal myocardial infarction; 4. fatal and non-fatal stroke. Secondary outcomes: 1. other cardiovascular events, including atrial fibrillation, arrhythmias, angina, revascularization, etc.; 2. total cholesterol (TC), triglycerides (TG), high-density lipoprotein (HDL), low-density lipoprotein (LDL); 3. systolic blood pressure (SBP) and diastolic blood pressure (DBP).

### Study selection

Two reviewers (BR Cheng and JQ Chen) independently screened all titles and abstracts of the records. Full texts of potentially eligible studies were retrieved for further identification according to the eligibility criteria. Any uncertainty or discrepancy was resolved by discussion. We used Excel for recording decisions.

### Data extraction

Two reviewers (BR Cheng and JQ Chen) independently extracted data following a predesigned data form using Excel (version Microsoft Excel 2016). Data were checked by an additional reviewer (XW Zhang). Disagreements were resolved by discussion.

### Risk of bias assessment

Since only RCTs were included, the risk of bias was assessed through Cochrane's risk-of-bias Tool for Randomized Trials (RoB 2) [24]. Two review authors (BR Cheng and QY Gao) independently assessed the risk of bias. Disagreements were resolved by discussion with an additional reviewer (XW Zhang). The risk of bias are assessed through the following five domains: (1) bias arising from the randomization process; (2) bias due to deviations from intended interventions; (3) bias due to missing outcome data; (4) bias in the measurement of the outcome; (5) bias in the selection of the reported result.

### Data analysis and synthesis

A narrative synthesis of the findings from the included studies will be provided. We worked with the data within a meta-analysis, through Review Manager 5.3. Heterogeneity related to the results of the studies was assessed using both the chi-square test and the $I^2$ statistic. If data were sufficiently homogenous, we would pool the results using a fixed-effect model, with standardized mean differences for continuous outcomes (in our review referring to TC, TG, HDL, LDL, systolic blood pressure, and diastolic blood pressure) and risk ratios for binary outcomes (in our review referring to all-cause mortality, cardiovascular mortality, fatal and non-fatal myocardial infarction, fatal and non-fatal stroke, and other cardiovascular events), and calculate 95% confidence intervals and two-sided P values for each outcome. We will provide summaries of intervention effects for each study by calculating risk ratios (for dichotomous outcomes) or standardized mean differences (for continuous outcomes). We considered an $I^2$ value greater than 50% indicative of substantial heterogeneity. If there were high heterogeneity across included studies, we would use a random effect model or only provide a narrative synthesis of the findings from the included studies, structured around the type of intervention, target population characteristics, type of outcome, and intervention content.

## Results

### Study selection

Database searches initially identified 6279 records in languages including English, Chinese, German, Russian, Turkish and Spanish. The last search date was July 22[nd], 2021. After removing duplications, 4312 articles were screened by their titles and abstracts. 166 full-texts were assessed for eligibility, but we failed to find 5 of them, as 3 were anonymous and the other 2, written in Russian and Turkish respectively, had no available resources online. A flowchart (Fig 1) with the number of included studies at each step was established, including reasons for excluding studies. 21 studies were included in qualitative and 20 in the quantitative synthesis.

### Study characteristics

Table 1 presents the characteristics of the studies included through the systematic review process. Except for basic study information, we also paid attention to if patients with cardiovascular risk were excluded and if aspirin was allowed to use for cardiovascular protection, which may affect the result of adverse events.

### Risk of bias of individual studies

Fig 2 and Fig 3 include a summary of the risk of bias assessed for each study included in this systematic review. Overall, most included studies were of low or unclear risks. Six studies were of high risk, which was mostly contributed by lack of specific randomization process or high withdrawal rates due to adverse events.

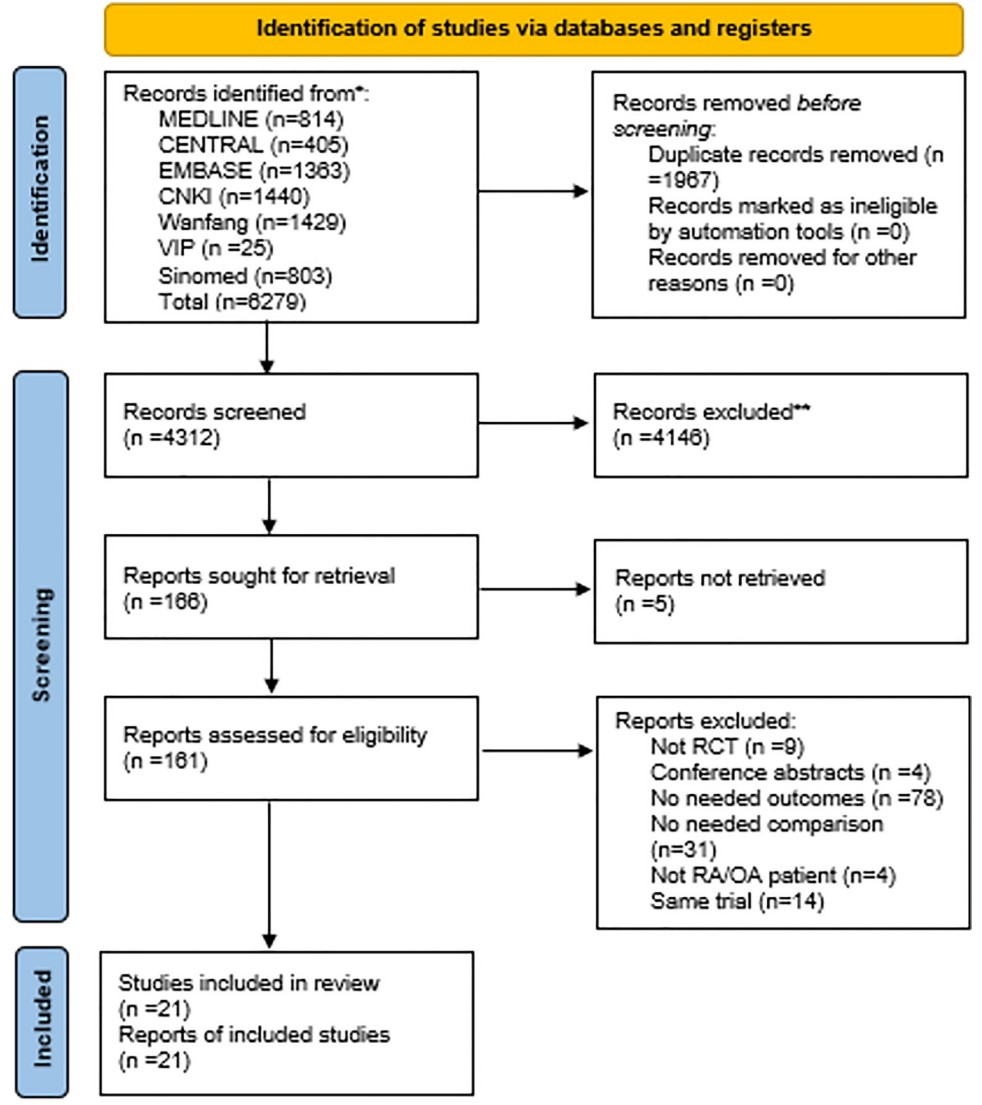

**Fig 1. Flowchart of the study selection process.**

## Primary outcomes

**All-cause mortality.** A total of 9 trials reported this outcome. The risk of all-cause mortality was decreased statistically significantly in celecoxib groups, compared with nsNSAIDs groups (RR 0.81, 95% CI 0.66–0.98, $I^2$ = 21%). As for celecoxib versus placebo, four trials reported all-cause mortality cases, which showed no significant differences between the two treatments (RR 0.92, 95% CI 0.26–3.27, $I^2$ = 0%).

**Cardiovascular mortality.** The cardiovascular mortality rate of the celecoxib groups was also lower than nsNSAIDs groups (RR 0.75, 95% CI 0.57–0.99, $I^2$ = 0%). When compared with placebo, the risk was not significantly different (RR 3.02, 95% CI 0.36–25.27, $I^2$ = 0%).

**Myocardial infarction.** The risk of myocardial infarction was not significant compared with both nsNSAIDs (RR 1.08, 95% CI = 0.88–1.33) and placebo (RR 1.87, 95% CI = 0.39–8.90) In the 4 trials which included placebo groups, the rate of myocardial infarction was zero, which could affect the result.

**Table 1. Characteristics of randomized controlled trials included in the qualitative synthesis.**

| Study | Condition | Total Sample Size | Comparators | Celecoxib Doses and Frequency | Duration |
|---|---|---|---|---|---|
| Bingham 2007 (Trial 1) [25] | OA | 599 | Placebo | 200mg Qd | 26 weeks |
| Bingham 2007 (Trial 2) [25] | OA | 608 | Placebo | 200mg Qd | 26 weeks |
| Birbara 2006 (Trial 1) [26] | OA | 395 | Placebo | 200mg Qd | 6 weeks |
| Birbara 2006 (Trial 2) [26] | OA | 413 | Placebo | 200mg Qd | 6 weeks |
| Cannon 2008 [27] | OA | 433 | Placebo, Ibuprofen | 200mg Bid | 12 weeks |
| Clegg 2006 (GAIT) [28] | OA | 1583 | Placebo | 200mg/day | 24 weeks |
| Conaghan 2012 [29] | OA | 1399 | Placebo | 100mg/day | 12 weeks |
| Cryer 2012 (PROBE) [30] | OA | 8067 | (1) | (2) | 6 months |
| Dahlberg 2009 [31] | OA | 925 | Diclofenac | 200mg Qd | 1 year |
| Essex 2012 [32] | OA | 589 | Naproxen | 200mg Qd | 6 months |
| Gibofsky 2003 [33] | OA | 477 | Placebo | 200mg/day | 6 weeks |
| Hawel 2003 [34] | OA | 148 | Dexiprofen | 100mg/day | 15 days |
| MacDonald 2017 (SCOT) [35] | OA&RA | 7297 | Diclofenac, Ibuprofen and Naproxen | 169.8±80.6mg/day | (3) |
| McKenna 2001 [36] | OA | 182 | Placebo | 200mg Qd | 6 weeks |
| Nissen 2016 (PRECISION) [37] | OA&RA | 24081 | Naproxen and Ibuprofen | 209±37mg/day | 20 months (4) |
| Sampalis 2012 [38] | OA | 60 | Placebo | 200mg/day | 90 days |
| Sawitzke 2010 [39] | OA | 662 | Placebo | 200mg/day | 24 weeks |
| Schnitzer 2011 [40] | OA | 1262 | Placebo | 200mg Qd | 13 weeks |
| Simon 1999 [41] | RA | 1149 | Placebo and Naproxen | 100mg, 200mg, or 400mg Bid | 12 weeks |
| Smugar 2006 (Trial 1) [42] | OA | 1521 | Placebo | 200mg Qd | 6 weeks |
| Smugar 2006 (Trial 2) [42] | OA | 1082 | Placebo | 200mg Qd | 6 weeks |
| White 2002 (CLASS) [43] | OA&RA | 7968 | Diclofenac and Ibuprofen | 100 mg Bid in patients with OA and up to 200 mg Bid in patients with RA | 13 weeks |
| Williams 2001 [44] | OA | 718 | Placebo | 100mg Bid or 200mg Qd | 6 weeks |
| Wittenberg 2006 [45] | OA | 364 | Placebo | 200mg Bid | 1 week |

(1): The nsNSAID in the comparator group was any nsNSAID of the investigator's choice, prescribed within the dosages allowed in the United States package insert.

(2): Celecoxib dosage could be adjusted within the United States prescribing guidelines.

(3): The median intention-to-treat follow-up for the primary outcome was 3.0 years (maximum 6.3 years, total 22 600 person-years)

(4): The mean duration of treatment was 20.3±16.0 months for all patients.

OA: Osteoarthritis; RA: Rheumatoid Arthritis; Qd: once per day; Bid: twice per day; GAIT: Glucosamine/chondroitin Arthritis Intervention Trial; PROBE: Prospective, Randomized, Open-label, Blinded Endpoint; SCOT: Standard care vs. Celecoxib Outcome Trial; PRECISION: Prospective Randomized Evaluation of Celecoxib Integrated Safety versus Ibuprofen Or Naproxen; mg: milligram; nsNSAID: non-selective Non-Steroidal Anti-Inflammatory Drug.

**Stroke.** The risk of stroke in the celecoxib group was also not significant compared with both nsNSAIDs and placebo (RR 0.94, 0.96, 95% CI = 0.71–1.24, 0.13–6.92).

## Secondary outcomes

Secondary outcomes included dichotomous and continuous data. Dichotomous data were the numbers of atrial fibrillation, arrhythmias, angina, revascularization, and heart failure, of which some were only reported in single studies (atrial fibrillation, celecoxib versus nsNSAIDs and placebo; arrhythmias, celecoxib versus nsNSAIDs), some in no studies (arrhythmias, celecoxib versus placebo; revascularization, celecoxib versus placebo; heart failure, celecoxib versus

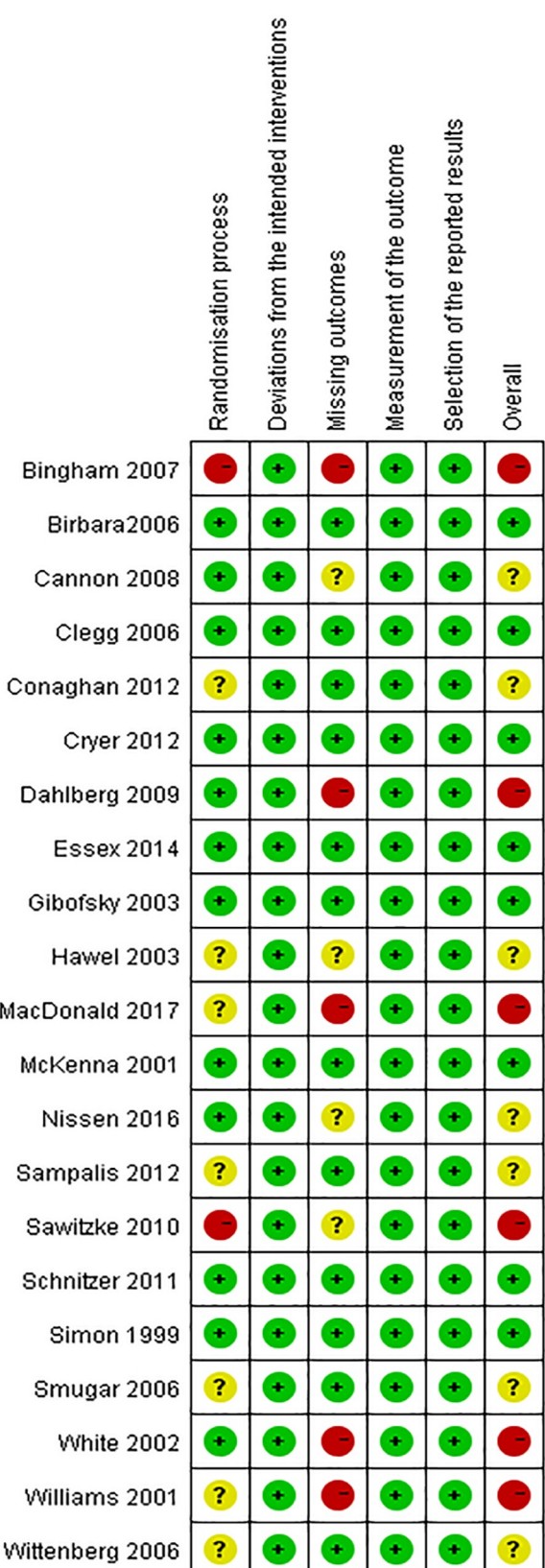

**Fig 2. Risk of bias summary: A review of authors' judgements about each risk of bias item for each included study.**

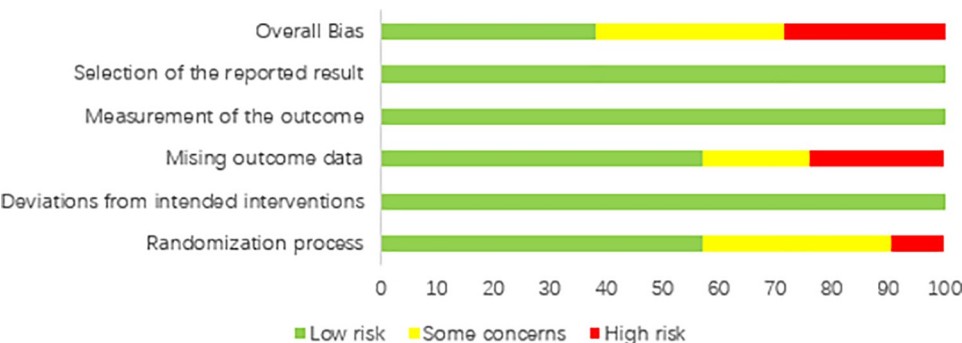

**Fig 3. Risk of bias graph: A review of authors' judgements about each risk of bias item presented as percentages across all included studies.**

placebo), and none of the results was statistically significant. More details are shown in Fig 4 and Fig 5.

Continuous data included the levels of total cholesterol, triglycerides, high-density lipoprotein, low-density lipoprotein, systolic blood pressure, and diastolic blood pressure. Blood pressures were reported in 3 studies: Bingham 2007, Sampalis 2012, and Simon 1999, all of which compared the celecoxib and placebo groups. Sampalis 2012 did not present any data for meta-analysis, but in their study, no significant change of both systolic and diastolic blood pressures was seen in the celecoxib or placebo groups. Bingham 2007 reported the mean changes of systolic and diastolic blood pressures before and after the study, while Simon 1999 reported the average systolic and diastolic blood pressures before and after the study, so their data couldn't be pooled, but their conclusions are the same: no differences were found between the celecoxib and placebo groups. As for the lipids and lipoproteins, only LDL-C levels were reported, in only one study [27], where the effects of celecoxib, placebo, and ibuprofen on LDL-C were comparable in patients with osteoarthritis.

## Discussion

Overall, celecoxib does not significantly increase cardiovascular events compared with placebo and slightly decreases the risk of all-cause mortality and cardiovascular mortality compared with nsNSAIDs, which may prove it safe when used on rheumatoid arthritis and osteoarthritis patients. Our conclusion was basically in line with the result of the PRECISION trial, in which celecoxib was found to be non-inferior to naproxen or ibuprofen for cardiovascular death, nonfatal myocardial infarction, or nonfatal stroke [37], and also in line with a prospective observational study in Japan assessing the cardiovascular risk between celecoxib and nsNSAIDs in patients with rheumatoid arthritis and osteoarthritis [46].

The all-cause mortality rate of celecoxib decreases significantly than that of nsNSAIDs but not that of placebo. However, since these causes include car accidents, suicide, and other irrelevant causes, it is not of much importance to this study.

The cardiovascular mortality rate comparing celecoxib and nsNSAIDs was reported in five studies, three of them excluded patients with cardiovascular risk (Cryer 2012, Dahlberg 2009 and MacDonald 2017), and aspirin was allowed in three studies (Dahlberg 2009, Nissen 2016 and White 2002), which may decrease the number of adverse cardiovascular events. In the secondary analysis of the CLASS study, the subgroup of patients not taking aspirin was analyzed separately and no differences were observed between celecoxib and the nsNSAID groups in terms of myocardial infarction or stroke with the exception that celecoxib was associated with a lower incidence of sudden cardiac death than diclofenac [47].

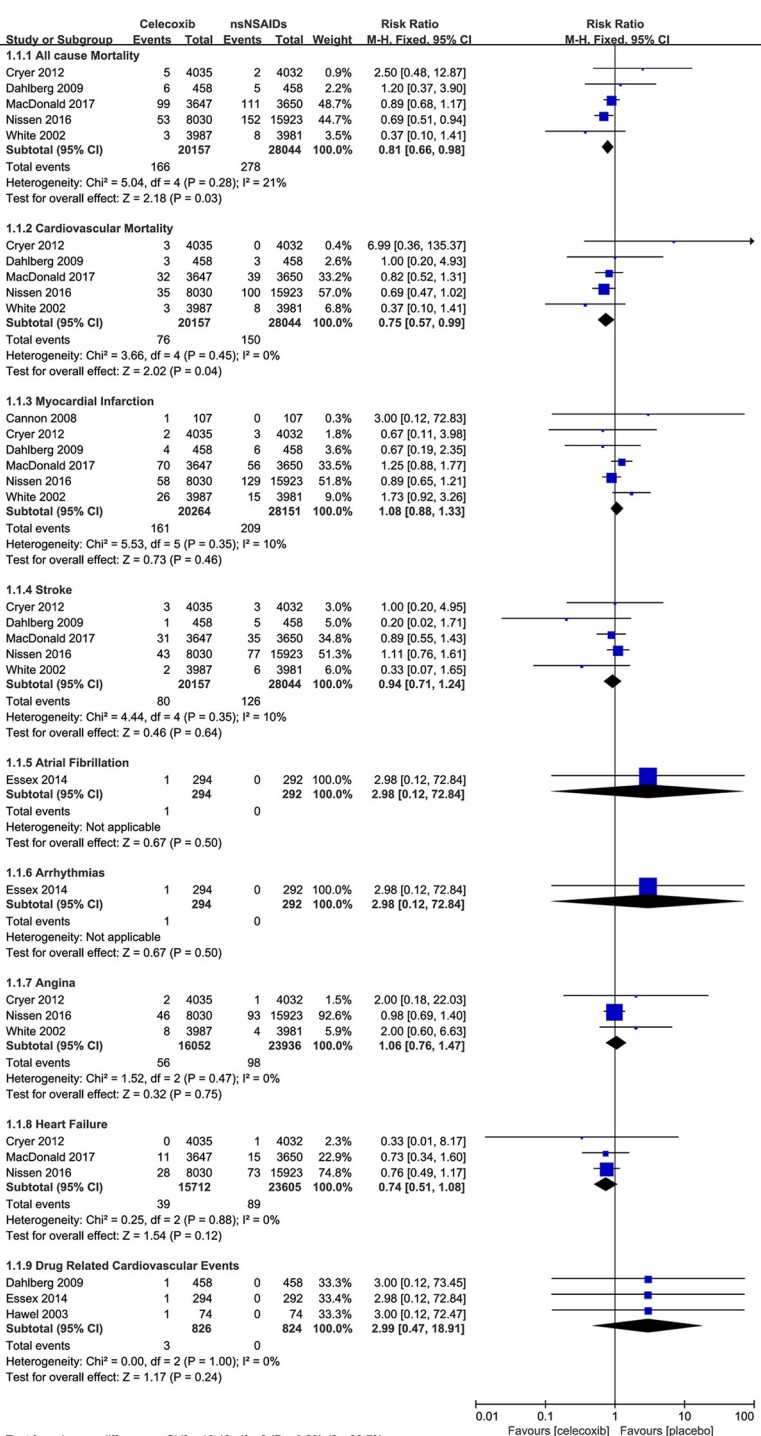

**Fig 4. Relative risk of dichotomous outcomes for celecoxib versus nsNSAIDs.**

As only 2 studies (Schnitzer 2011 and Williams 2001) were included in this primary outcome analysis, although it reported no differences in cardiovascular mortality rate between celecoxib and placebo, it is not considered to be persuasive. In a study on the cardiovascular safety of NSAIDs (patients' conditions unlimited), celecoxib was found to be related to a

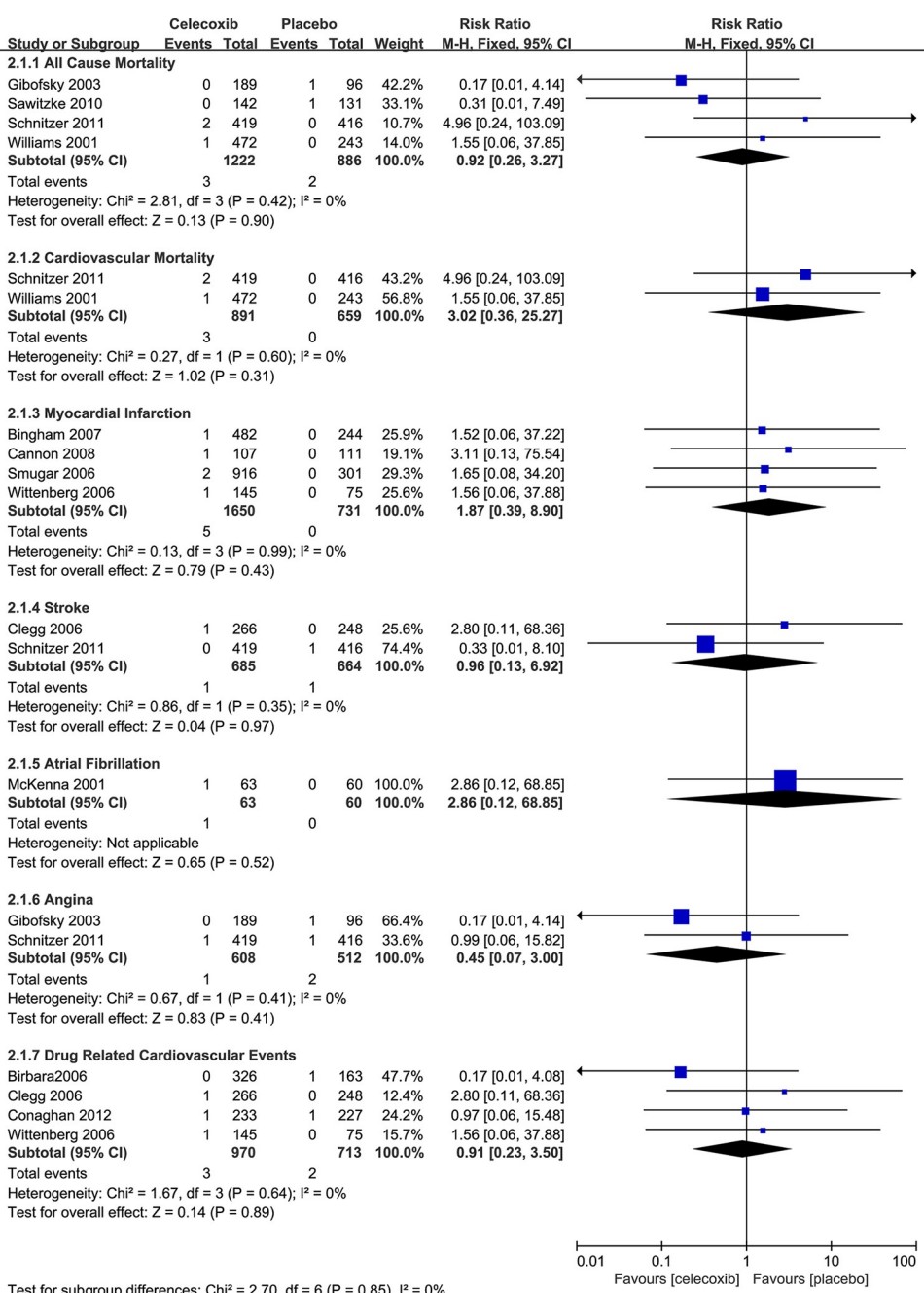

**Fig 5. Relative risk of dichotomous outcomes for celecoxib versus placebo.**

higher risk of cardiovascular death, as well as myocardial infarction, stroke, death from any cause, and APTC composite outcome, but none of them was statistically significant [48]. Our results did not present any significant differences either.

The use of celecoxib on blood pressure seems to be safe. Sampalis et al. found that both celecoxib and placebo did not influence systolic blood pressure or diastolic blood pressure. Curtis E et al. also found no significant increase in hypertension in celecoxib and etoricoxib compared with placebo, albeit an increase in the risk of heart failure and edema [17].

But it remains uncertain if celecoxib is more beneficial in patients with established cardio-vascular risk (stroke history, hypertension, etc.). As the mechanism of elevated cardiovascular risk in RA and OA remains unveiled, there has been uncertainty about the nature and magni-tude of these risks. Vitamin B-6 could be a possible factor, as its metabolism can be impaired by clinical use of cyclooxygenase inhibitors [49], and it is associated with higher cardiovascular risks [50, 51]. Still, we could not know if the cardiovascular safety profile of celecoxib is the same in all patients with different cardiovascular risks. Whether it is more beneficial or more dangerous in patients with specific cardiovascular risk is still nowhere to know.

What's more, there were discrepancies between real-world clinical practice and randomized clinical trials. Studies showed that the use of coxibs tended to be shorter, more variable, and at lower doses in clinical practice than in some clinical trials. As NSAIDs are always used as pain-killers, so not every patient needs to take them every day at a specific dose. Therefore, most patients probably have been exposed to doses and duration sufficient to detect an increased cardiovascular risk [52]. This provided a basis for reconciling the apparent discrepancies between these trials and explained the findings from most non-experimental epidemiologic studies, which showed no increased risk with celecoxib, irrespective of dose [53].

Meanwhile, effective biomarkers to predict the cardiovascular risk associated with the use of anti-inflammatory drugs for RA and OA are in need to better provide cardiovascular pro-tective prescriptions. NT-proBNP may be a useful marker for anticipating cardiovascular risk in OA patients [54].

Also, since valdecoxib is more selective for COX-2 than celecoxib in vitro, it is related to higher cardiovascular risk. However, there may well be some patients in whom celecoxib is the more selective inhibitor in vivo [55]. Therefore, better indications cannot be made until the mechanisms behind RA and OA patients' elevated cardiovascular risk are unveiled.

Last, our research had some limitations. First, some of the studies included in this research were not intended to assess the cardiovascular risk of celecoxib, even though safety profiles were reported. Second, as the adverse cardiovascular events rate is relatively small, many stud-ies included in full-text screening did not report any cardiovascular events. Some other studies that reported cardiovascular events were of small samples or of short duration, which may dilute the cardiovascular adverse effect of drugs. Also, publication bias couldn't have been assessed properly due to the small number (most of them fewer than 5) of trials included in each meta-analysis, which added uncertainty to our conclusion.

However, despite a growing understanding of the mechanisms of RA and their interplay with cardiovascular risk factors, it also remains unclear the relationship between arthritis and increase cardiovascular events [56]. Discovering the mechanism is essential to pursue the spe-cific treatment of this common comorbidity.

## Conclusion

This meta-analysis found celecoxib does not increase the cardiovascular risk compared with common nsNSAIDs and placebo, and therefore confirmed celecoxib can be safely prescribed at approved doses in RA or OA patients as a first-line NSAID. But safety data are needed con-sidering long duration or high dose usage. The result remains uncertain in patients treated with aspirin and patients with established cardiovascular diseases.

## Supporting information

**S1 Appendix. PRISMA 2020 flow diagram.**
(DOCX)

**S2 Appendix. PRISMA 2020 checklist.**
(DOCX)

**S3 Appendix. PROSPERO protocol.**
(PDF)

**S4 Appendix. Search strategy.**
(DOCX)

**S5 Appendix. Data extraction of study characteristics.**
(XLS)

**S6 Appendix. Data extraction of outcomes.**
(XLS)

**S7 Appendix. RoB assessment.**
(XLSM)

## Author Contributions

**Formal analysis:** Bai-Ru Cheng, Jia-Qi Chen, Qin-Yang Gao.

**Funding acquisition:** Jian-Ping Liu.

**Methodology:** Bai-Ru Cheng, Jia-Qi Chen, Xiao-Wen Zhang, Wei-Hong Li, Li-Jiao Yan, Jing-Li Xing, Jian-Ping Liu.

**Project administration:** Bai-Ru Cheng, Jia-Qi Chen.

**Software:** Bai-Ru Cheng, Yu-Qiao Zhang, Chang-Jiang Wu.

**Supervision:** Bai-Ru Cheng, Xiao-Wen Zhang, Jian-Ping Liu.

**Writing – original draft:** Bai-Ru Cheng.

**Writing – review & editing:** Bai-Ru Cheng, Xiao-Wen Zhang, Jian-Ping Liu.

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
