## [Decision Letter · Decision Letter 0]

13 Sep 2021

PONE-D-21-23851Cardiovascular Safety of Celecoxib in Rheumatoid Arthritis and Osteoarthritis Patients: A Systematic Review and Meta-AnalysisPLOS ONE

Dear Dr. Liu,

Thank you for submitting your manuscript to PLOS ONE. After careful consideration, we feel that it has merit but does not fully meet PLOS ONE’s publication criteria as it currently stands. Therefore, we invite you to submit a revised version of the manuscript that addresses the points raised during the review process.

The reviewers made several important points that should be addressed, particularly the statistics require attention. Please consider to to seek advice from a biostatistician (preferably also including him/her as an author.

Furthermore, the English needs substantial improvement.

We look forward to receiving your revised manuscript.

Kind regards,

Michael Nurmohamed, MD, PhD

Academic Editor

PLOS ONE

Journal Requirements:

“This review was supported by the National Natural Science Foundation project (No. 81830115), and partially supported by the NCCIH grant (AT001293 with sub award No. 020468C).”

5. Please ensure that you refer to Figure 4, 5, 6, 7 and 8 in your text as, if accepted, production will need this reference to link the reader to the figure.

Reviewers' comments:

Reviewer's Responses to Questions

**Comments to the Author**

1. Is the manuscript technically sound, and do the data support the conclusions?

Reviewer #1: Yes

Reviewer #2: Partly

2. Has the statistical analysis been performed appropriately and rigorously? 

Reviewer #1: Yes

Reviewer #2: No

3. Have the authors made all data underlying the findings in their manuscript fully available?

Reviewer #1: Yes

Reviewer #2: Yes

4. Is the manuscript presented in an intelligible fashion and written in standard English?

Reviewer #1: No

Reviewer #2: No

5. Review Comments to the Author

Reviewer #1: This is a timely manuscript which aims to provide some higher level of evidence in a relative arguable subject, namely the overall and in particular the cardiovascular safety profile of celecoxib and NSAIDs in patients with OA and/or RA. The authors performed a systematic review and a meta-analysis of available literature and found that celecoxib has a better CV profile than other NSAIDs in terms of all-cause mortality and CV mortality, whereas no differences have been observed for myocardial infarction and CVA. Similarly, no differences have been shown with respect to all analyzed outcomes when celecoxib has been compared to placebo. Some points could be made that should be addressed in order to improve the manuscript:

1. English language: please have the manuscript checked by a native English speaker or someone with highest qualifications in English language.

2. Methodologic issues/statistical issues: the study of Nissen (2016) and MacDonald (2017) are the largest studies and both of them provide data supporting a neutral/beneficial effect of celecoxib with respect to all-cause mortality and CV mortality. Likewise, it is of little surprise that the meta-analysis imbedding this two studies would come to the same conclusion, only now reaching statistical significance. My point is that we should still be caution when interpreting the results, because even this is a meta-analysis, it seems that these two studies are actually the ones determining the final outcome. Could the authors discuss this point?

3. Clinical relevance: from the clinician point of view it is of relevance to know if it is possible to safely prescribe NSAIDs to a patient, and which one fits the best with his clinical profile. Among the studies included in this meta-analysis a few investigated RA patients. Is it possible to perform separate analysis for the OA and RA patients respectively? While NSAIDs are often prescribed as painkillers in OA (where paracetamol represents an equally good alternative, given the low-inflammatory status in OA), their prescription in RA can sometimes be determined by an increasing of disease activity, thus working more as anti-inflammatory drugs, decreasing inflammation and consequently its detrimental cardiovascular effects. Therefore one may hypothesize that the same drug (celecoxib) might have different CV impact in RA and OA respectively because of different biological effects (more painkilling (OA) vs. more anti-inflammatory (RA)).

It would therefore be very useful if the authors would provide the readers with an answer to the following questions (based on their results):

a) Can I safely prescribe celecoxib in an OA patient? As a first NSAID? What if the patient has a history of CV events?

b) Can I safely prescribe celecoxib to an RA patient? As a first NSAID? What if the patient has a history of CV events?

Thank you.

Reviewer #2: Liu et al. have written a systematic review and meta-analysis about the use of celecoxib and its cardiovascular safety in patients with rheumatoid arthritis and osteoarthritis. This is a relevant topic for daily clinical practice. However, I have several concerns about the manuscript.

1. In the introduction, on page 2, line 47 and 48, the authors write about coronary atherosclerosis and non-fatal myocardial infarction. I think this sentence should be refrased. Now it reads as if coronary atherosclerosis and a myocardial infarction are different entities. It could be refrased to "particularly (fatal and non-fatal) myocardial infarction due to coronary atherosclerosis".

2. The authors write that there are no specific treatment strategies for CV disease in RA patients. However, the EULAR recommendations state that there is evidence (from mainly observational studies) that an optimal control of disease activity and treatment of CVD risk factors reduces the CVD risk in patients with an inflammatory joint disease.

3. Regarding the statistical analysis I would like to advise to discuss the methods with a statistician. Studies with zero events were included in the analyses. I wonder if this is the correct way to do these analyses. I wonder if you can conclude that all-cause mortality was decreased in the celecoxib group when there are 0-2 events in the studies included.

4. I could not find which non-selective NSAIDs the authors included in their search and comparison with celecoxib. I think this is relevant and should be added to the manuscript.

5. In general, I would advise to ask a native English speaker to review the manuscript.

6. PLOS authors have the option to publish the peer review history of their article (what does this mean?). If published, this will include your full peer review and any attached files.

Reviewer #1: No

Reviewer #2: No

---

## [Author Response · Author response to Decision Letter 0]

24 Oct 2021

Reviewer #1: This is a timely manuscript which aims to provide some higher level of evidence in a relative arguable subject, namely the overall and in particular the cardiovascular safety profile of celecoxib and NSAIDs in patients with OA and/or RA. The authors performed a systematic review and a meta-analysis of available literature and found that celecoxib has a better CV profile than other NSAIDs in terms of all-cause mortality and CV mortality, whereas no differences have been observed for myocardial infarction and CVA. Similarly, no differences have been shown with respect to all analyzed outcomes when celecoxib has been compared to placebo. Some points could be made that should be addressed in order to improve the manuscript:

1. English language: please have the manuscript checked by a native English speaker or someone with highest qualifications in English language.

Response: We apologize for the insufficient quality of the original manuscript. This issue has been highly recognized by our team, and multiple times of revisions with great scrutiny have been performed. Some grammatical errors and expressions have been checked. We hope we have presented a better version. Thank you. 

2. Methodologic issues/statistical issues: the study of Nissen (2016) and MacDonald (2017) are the largest studies and both of them provide data supporting a neutral/beneficial effect of celecoxib with respect to all-cause mortality and CV mortality. Likewise, it is of little surprise that the meta-analysis imbedding this two studies would come to the same conclusion, only now reaching statistical significance. My point is that we should still be caution when interpreting the results, because even this is a meta-analysis, it seems that these two studies are actually the ones determining the final outcome. Could the authors discuss this point? 

Response: Thank you for your question. If the amount of included literature meta-analysis is small, such as two studies as you mentioned, there is often a serious bias. In principle, we can see whether there is publication bias through the funnel chart for more than 9 documents. Publication bias is an important aspect which should be considered because in many cases, articles with negative results are less likely to be published. As for other biases, we can only find out based on the evaluation of the quality of the literature (criteria and methodology). The available literature should be included in the study according to the PRISMA guidelines.

3. Clinical relevance: from the clinician point of view it is of relevance to know if it is possible to safely prescribe NSAIDs to a patient, and which one fits the best with his clinical profile. Among the studies included in this meta-analysis a few investigated RA patients. Is it possible to perform separate analysis for the OA and RA patients respectively? While NSAIDs are often prescribed as painkillers in OA (where paracetamol represents an equally good alternative, given the low-inflammatory status in OA), their prescription in RA can sometimes be determined by an increasing of disease activity, thus working more as anti-inflammatory drugs, decreasing inflammation and consequently its detrimental cardiovascular effects. Therefore one may hypothesize that the same drug (celecoxib) might have different CV impact in RA and OA respectively because of different biological effects (more painkilling (OA) vs. more anti-inflammatory (RA)).

It would therefore be very useful if the authors would provide the readers with an answer to the following questions (based on their results):

a) Can I safely prescribe celecoxib in an OA patient? As a first NSAID? What if the patient has a history of CV events?

b) Can I safely prescribe celecoxib to an RA patient? As a first NSAID? What if the patient has a history of CV events?

Thank you.

Response: Thank you for your question. Actually, before the submission, we tried to do this subgroup analysis you mentioned but failed because only 1 of the 24 trials included in our systematic review recruited only RA patients. 3 trials recruited both OA and RA patients, but data of OA and RA subgroups were not provided. Therefore, it is difficult to do this subgroup analysis. But based on the evidence we have so far, we consider it safe to prescribe celecoxib at an approved dose - 100, 200, and 400mg per day - in an OA patient, as a recommended NSAID. But we dare not say “a first NSAID” because naproxen also showed great safety and no statistical differences have been found between them in the study of Essex (2012)[1], MacDonald (2017)[2], Nissen (2016) [3] and Simon (1999)[4]. As for RA patients, although the participants were fewer, we still think it’s OK to draw the same conclusion because 1) any enhancement of CV risk with celecoxib or nsNSAID is modest[2]; and 2) trials with OA&RA patients presented the same results as those of trials with only OA patients. However, no subgroup analysis was reported considering patients with/without a history of CV events, so we cannot give a sound conclusion so far. We have added some related contents in the CONCLUSION.

Reviewer #2: Liu et al. have written a systematic review and meta-analysis about the use of celecoxib and its cardiovascular safety in patients with rheumatoid arthritis and osteoarthritis. This is a relevant topic for daily clinical practice. However, I have several concerns about the manuscript.

1. In the introduction, on page 2, line 47 and 48, the authors write about coronary atherosclerosis and non-fatal myocardial infarction. I think this sentence should be refrased. Now it reads as if coronary atherosclerosis and a myocardial infarction are different entities. It could be refrased to "particularly (fatal and non-fatal) myocardial infarction due to coronary atherosclerosis".

Response: Thank you for your advice and we apologize for the insufficient quality of the original manuscript. We value your opinion very much and have rephrased it as you suggested. 

2. The authors write that there are no specific treatment strategies for CV disease in RA patients. However, the EULAR recommendations state that there is evidence (from mainly observational studies) that an optimal control of disease activity and treatment of CVD risk factors reduces the CVD risk in patients with an inflammatory joint disease.

Response: Thank you for your comments. Our purpose by saying “there were no specific treatment strategies” was that the relationship between RA and its raised CV risk remains unclear. Therefore, no specific treatments could have been found. We deeply agree with the EULAR recommendations that controlling disease activity and treatment of the risk factors reduces the CVD risk. To minimize misunderstanding, we deleted related lines.

3. Regarding the statistical analysis I would like to advise to discuss the methods with a statistician. Studies with zero events were included in the analyses. I wonder if this is the correct way to do these analyses. I wonder if you can conclude that all-cause mortality was decreased in the celecoxib group when there are 0-2 events in the studies included. 

Response: Thanks for the advice. We discussed the methods with a statistician (Jingli Xing), and she found no mistakes in them. In a meta-analysis, when we want to calculate the rate of a specific event, the exact number of the event should be recorded, even though sometimes this event did not happen - meaning the rate is 0%. In Cochrane Handbook for Systematic Reviews of Interventions Version 6.2, Chapter 10.4.4.2: Studies with no events in either arm, it wrote: “Whilst the results of risk difference meta-analyses will be affected by non-reporting of outcomes with no events, odds and risk ratio based methods naturally exclude these data whether or not they are published, and are therefore unaffected”[5]. Therefore, we think it is appropriate to include studies with zero events.

4. I could not find which non-selective NSAIDs the authors included in their search and comparison with celecoxib. I think this is relevant and should be added to the manuscript.

Response: Thanks for your question. The details about which non-selective NSAIDs, if any, were included in each trial are concluded in “Table 1 Characteristics of randomized controlled trials included in qualitative synthesis”, which contains ibuprofen, diclofenac, naproxen, and dexiprofen. In the PROBE trial, the nsNSAID in the comparator group was any nsNSAID of the investigator’s choice, prescribed within the dosages allowed in the United States package insert. In most of the trials, celecoxib was compared with placebo. As the incidence of events decreases, the logarithmic conversion, logit conversion, and arcsine conversion methods can still maintain good performance.

5. In general, I would advise to ask a native English speaker to review the manuscript.

Response: We apologize for the insufficient quality of the original manuscript. This issue has been highly recognized by our team, and multiple times of revisions with great scrutiny have been performed this time. Some grammatical errors and expressions have been checked. We hope we have presented a better manuscript. Thank you. 

References:

[1] ESSEX M N, BHADRA P, SANDS G H. Efficacy and tolerability of celecoxib versus naproxen in patients with osteoarthritis of the knee: a randomized, double-blind, double-dummy trial [J]. J Int Med Res, 2012, 40(4): 1357-70.

[2] MACDONALD T M, HAWKEY C J, FORD I, et al. Randomized trial of switching from prescribed non-selective non-steroidal anti-inflammatory drugs to prescribed celecoxib: the Standard care vs. Celecoxib Outcome Trial (SCOT) [J]. Eur Heart J, 2017, 38(23): 1843-50.

[3] NISSEN S E, YEOMANS N D, SOLOMON D H, et al. Cardiovascular Safety of Celecoxib, Naproxen, or Ibuprofen for Arthritis [J]. N Engl J Med, 2016, 375(26): 2519-29.

[4] SIMON L S, WEAVER A L, GRAHAM D Y, et al. Anti-inflammatory and upper gastrointestinal effects of celecoxib in rheumatoid arthritis: a randomized controlled trial [J]. JAMA, 1999, 282(20): 1921‐8.

[5] Deeks JJ, Higgins JPT, Altman DG (editors). Chapter 10: Analysing data and undertaking meta-analyses. In: Higgins JPT, Thomas J, Chandler J, Cumpston M, Li T, Page MJ, Welch VA (editors). Cochrane Handbook for Systematic Reviews of Interventions version 6.2 (updated February 2021). Cochrane, 2021. Available from www.training.cochrane.org/handbook.

---

## [Decision Letter · Decision Letter 1]

22 Nov 2021

PONE-D-21-23851R1Cardiovascular Safety of Celecoxib in Rheumatoid Arthritis and Osteoarthritis Patients: A Systematic Review and Meta-AnalysisPLOS ONE

Dear Dr. Jian-Ping Liu .

Thank you for submitting your manuscript to PLOS ONE. After careful consideration, we feel that it has merit but does not fully meet PLOS ONE’s publication criteria as it currently stands. Therefore, we invite you to submit a revised version of the manuscript that addresses the points raised during the review process.

 There are are some remaining methodological issues, raised by reviewer 1, that need to be addressed

We look forward to receiving your revised manuscript.

Kind regards,

Michael Nurmohamed, MD, PhD

Academic Editor

PLOS ONE

Journal Requirements:

Reviewers' comments:

Reviewer's Responses to Questions

**Comments to the Author**

1. If the authors have adequately addressed your comments raised in a previous round of review and you feel that this manuscript is now acceptable for publication, you may indicate that here to bypass the “Comments to the Author” section, enter your conflict of interest statement in the “Confidential to Editor” section, and submit your "Accept" recommendation.

Reviewer #1: (No Response)

Reviewer #2: All comments have been addressed

2. Is the manuscript technically sound, and do the data support the conclusions?

Reviewer #1: Yes

Reviewer #2: Yes

3. Has the statistical analysis been performed appropriately and rigorously? 

Reviewer #1: I Don't Know

Reviewer #2: Yes

4. Have the authors made all data underlying the findings in their manuscript fully available?

Reviewer #1: Yes

Reviewer #2: Yes

5. Is the manuscript presented in an intelligible fashion and written in standard English?

Reviewer #1: Yes

Reviewer #2: Yes

6. Review Comments to the Author

Reviewer #1: I would like to ask the authors to address in the Discussion the issue of publication bias, as a important limitation to their meta-analysis.

Secondly, I would like to know if there is an explanation for the fact that 11 studies mentioned in Table 1 (references no. 25,26, 28, 29, 32, 34, 36, 38, 41, 42, 45) can not be found among the studies presented in the figures 4 - 8. Is the data from these 11 studies excluded from the meta-analysis?

Finally, I would like to ask the authors to sharpen their aim of the study at the end of the introduction and also to address in their discussion or conclusion the gain of knowledge for the scientific community resulting from their meta-analysis. Thank you.

Reviewer #2: (No Response)

7. PLOS authors have the option to publish the peer review history of their article (what does this mean?). If published, this will include your full peer review and any attached files.

Reviewer #1: No

Reviewer #2: No

---

## [Author Response · Author response to Decision Letter 1]

24 Nov 2021

Reviewer #1: 

Q1: I would like to ask the authors to address in the Discussion the issue of publication bias, as an important limitation to their meta-analysis.

Response: Thanks for your advice. As the number of trials included in each meta-analysis was small (fewer than 10 trials), publication bias could not have been assessed properly, which makes it an important limitation to our work. We have addressed this issue in the Discussion (Line 275-276).

Q2: Secondly, I would like to know if there is an explanation for the fact that 11 studies mentioned in Table 1 (references no. 25,26, 28, 29, 32, 34, 36, 38, 41, 42, 45) can not be found among the studies presented in the figures 4 - 8. Is the data from these 11 studies excluded from the meta-analysis?

Response: Thanks for your question. Figures 4-8 presented part of the pooled results of the primary outcomes, which are all-cause mortality, cardiovascular mortality, myocardial infarction, and stroke. Some negative results were only shown in the text because we were trying to reduce the number of pictures. As for the secondary outcomes, which are atrial fibrillation, arrhythmias, angina, revascularization, and heart failure, there were no figures in our last edition of the manuscript because some were only reported in single studies, some in no studies, and none of the results was statistically significant. We value your opinion very much and agree with you that it’s improper that we did not give more details about the 11 studies you mentioned, so we combined all the results of dichotomous outcomes into 2 figures (figure 4 and 5 in our latest manuscript), where 19 of the 21 studies included can be found except Sampalis 2012[1] and Simon 1999[2] for following reasons: 1. Sampalis 2012 did not present any data capable for analysis, so we only described their results in the text; 2. Simon 1999 was the only study that presented blood pressures in the celecoxib group and placebo group before and after the study, so the results couldn’t be pooled and also couldn’t be shown in Fig 5. Relative risk of dichotomous outcomes for celecoxib versus placebo as they were continuous data. Therefore, we only described their findings in the text. 

Q3: Finally, I would like to ask the authors to sharpen their aim of the study at the end of the introduction and also to address in their discussion or conclusion the gain of knowledge for the scientific community resulting from their meta-analysis. Thank you.

Response: Thanks for your advice. We also think doing so helps readers to better find their take-home messages, so some changes were made in the text (Line 84-86, and Line294-295) to address the aim and findings clearer.

Thank you again for all your precious comments! They’ve been really helpful to us and this article.

References:

[1] SAMPALIS J S, BROWNELL L A. A randomized, double blind, placebo and active comparator controlled pilot study of UP446, a novel dual pathway inhibitor anti-inflammatory agent of botanical origin [J]. Nutr J, 2012, 11(21.

[2] SIMON L S, WEAVER A L, GRAHAM D Y, et al. Anti-inflammatory and upper gastrointestinal effects of celecoxib in rheumatoid arthritis: a randomized controlled trial [J]. JAMA, 1999, 282(20): 1921‐8.

---

## [Editor Report · Decision Letter 2]

26 Nov 2021

Cardiovascular Safety of Celecoxib in Rheumatoid Arthritis and Osteoarthritis Patients: A Systematic Review and Meta-Analysis

PONE-D-21-23851R2

Dear Dr. Jian-Ping Liu,

We’re pleased to inform you that your manuscript has been judged scientifically suitable for publication and will be formally accepted for publication once it meets all outstanding technical requirements.

Kind regards,

Michael Nurmohamed, MD, PhD

Academic Editor

PLOS ONE

---

## [Editor Report · Acceptance letter]

6 Dec 2021

PONE-D-21-23851R2 

Cardiovascular Safety of Celecoxib in Rheumatoid Arthritis and Osteoarthritis Patients: A Systematic Review and Meta-Analysis 

Dear Dr. Liu:

I'm pleased to inform you that your manuscript has been deemed suitable for publication in PLOS ONE. Congratulations! Your manuscript is now with our production department. 

Kind regards, 

on behalf of

Prof.Dr Michael Nurmohamed 

Academic Editor

PLOS ONE